# Isolation and Identification of Lactic Acid Bacteria Probiotic Culture Candidates for the Treatment of *Salmonella enterica* Serovar Enteritidis in Neonatal Turkey Poults

**DOI:** 10.3390/ani9090696

**Published:** 2019-09-17

**Authors:** Margarita A. Arreguin-Nava, Daniel Hernández-Patlán, Bruno Solis-Cruz, Juan D. Latorre, Xochitl Hernandez-Velasco, Guillermo Tellez, Saeed El-Ashram, Billy M. Hargis, Guillermo Tellez-Isaias

**Affiliations:** 1Eco-Bio LLC, Fayetteville, AR 72701, USA; marreguin@euxxis.bio; 2Laboratorio 5: LEDEFAR, Unidad de Investigación Multidisciplinaria, Facultad de Estudios Superiores Cuautitlán, Universidad Nacional Autónoma de México (UNAM), Cuautitlán Izcalli Estado de México 54714, Mexico; danielpatlan@comunidad.unam.mx (D.H.-P.); bruno_sc@comunidad.unam.mx (B.S.-C.); 3Department of Poultry Science, University of Arkansas, Fayetteville, AR 72701, USA; juandlatorre@gmail.com (J.D.L.); bhargis@uark.edu (B.M.H.); 4Departamento de Medicina y Zootecnia de Aves, Facultad de Medicina Veterinaria y Zootecnia, UNAM, Cd. de Mexico 04510, Mexico; xochitl_h@yahoo.com; 5School of Life Science and Engineering, Foshan University, Foshan 528231, Guangdong, China; saeed_elashram@yahoo.com; 6Faculty of Science, Kafrelsheikh University, Kafr el-Sheikh 33516, Egypt

**Keywords:** lactic acid bacteria, probiotic, turkey poult, microbiome, *Salmonella*

## Abstract

**Simple Summary:**

*Salmonella* spp. continues to be one of the most important foodborne bacterial pathogens. *S*. *enterica* serotype Enteritidis (SE) that emerged as an important human illness during the 1980s is currently one of the most common non-typhoidal *Salmonella* serotypes worldwide. Poultry and their products (eggs and meat) are considered as one of the most important source of SE infection in humans. Due to restrictions in the addition of antibiotics in the feed of animals intended for human consumption, alternatives to these antibiotics have been sought. Probiotics have shown to reduce infection in turkey poults. However, studies are lacking to show how these probiotics influence the intestinal microbiome as well as how this microbiome is related to a lower infection by *Salmonella*. In the present study the effect of a *Lactobacillus* spp.-based probiotic on SE colonization was evaluated in two separate experiments. In both trials, a significant reduction in the incidence and log_10_ cfu/g of SE were observed in poults treated with the probiotic when compared with control poults (*p* ≤ 0.05). Results showed that the application of this probiotic culture could reduce SE cecal colonization in day-of-hatch turkey poults, although further research is needed to elucidate the mechanism of this response.

**Abstract:**

The effect of *Lactobacillus* spp.-based probiotic candidates on *Salmonella* enterica serovar Enteritidis (SE) colonization was evaluated in two separate experiments. In each experiment, sixty-one day-of-hatch female turkey poults were obtained from a local hatchery. In both experiments, poults were challenged via oral gavage with 10^4^ cfu/poult of SE and randomly allocated to one of two groups (*n* = 30 poults): (1) the positive control group and (2) the probiotic treated group. Heated brooder batteries were used for housing each group separately and poults were allowed ad libitum access to water and unmedicated turkey starter feed. 1 h following the SE challenge, poults were treated with 10^6^ cfu/poult of probiotic culture via oral gavage or phosphate-buffered saline (PBS) to control groups. A total of 24 h post-treatment, poults were euthanized and the ceca and cecal tonsils from twenty poults were collected aseptically for SE recovery. In both trials, a significant reduction in the incidence and log_10_ cfu/g of SE were observed in poults treated with the probiotic when compared with control poults (*p* ≤ 0.05). The results of the present study suggest that the administration of this lactic acid-producing bacteria (LAB)-based probiotic 1 h after an SE challenge can be useful in reducing the cecal colonization of this pathogen in neonatal poults.

## 1. Introduction

A previous study reported nontyphoidal *Salmonella* spp., *Clostridium perfringens*, *Campylobacter* spp., and *Escherichia coli* as some of the most important foodborne bacterial pathogens in the U.S. [1]. Overall, health-related cost associated with the food borne illness from those pathogens was estimated to be around $51.0 and $77.7 billion based on a basic and enhanced model, respectively, as described earlier [1,2]. *S*. *enterica* serotype Enteritidis (SE) that emerged as an important human illness during the 1980s is currently one of the most common non-typhoidal *Salmonella* serotypes worldwide, especially in developed countries [3]. Poultry and their products (eggs and meat) are considered as one of the most important source of SE infection in humans. However, SE has also been isolated from non-poultry sources such as market hog carcasses, steer and heifer carcasses, cow and bull carcasses, and ground beef [4,5,6]. Due to the ban or restrictions on antibiotic growth promoters (AGPs), there are growing challenges for the poultry industry to cope with enteric pathogens such as *Salmonella*. This has created huge demands for finding alternatives to AGPs and, thus, several possible alternatives such as enzymes, organic acids, probiotics, prebiotics, etheric oils, and immunostimulants have been widely studied [7,8]. Hence, several studies have been conducted with the objective to reduce *Salmonella* spp. load in poultry and their products using various approaches such as antibodies, bacteriophages, probiotics, prebiotics, vaccines, and integrated farm management [9,10,11,12,13]. Although several approaches have already been studied, there is still a need to find better products that can work effectively with reproducible results. Over the last eighteen years, our laboratory has conducted extensive research to evaluate the antimicrobial capability of several lactic acid-producing bacteria (LAB) isolates from turkey origin, mainly against *Salmonella* spp. Some of these strains were selected to produce a commercial probiotic called FloraMax^®^-B11 (Pacific Vet Group, Fayetteville, AR, USA), which has been evaluated to prevent and treat *Salmonella* spp. infection and intestinal colonization in poultry [14,15,16,17]. Published commercial studies also showed that this probiotic culture reduced idiopathic diarrhea in commercial turkey [18] and increased performance and reduced costs of production [19,20,21,22]. In other studies, the administration of this probiotic 1 h after an SE challenge induced marked and rapid decreases between 12 and 24 h post-challenge [23]. Furthermore, the administration of FloraMax^®^-B11 after a 1 h post-*Salmonella* Heidelberg (SH) challenge practically eliminated the cecal colonization of SH [24]. These studies suggest some of the mechanisms that may be involved in the efficacy previously reported in laboratory and field conditions [25]. From the experience obtained during these years of research, in the present study, we evaluated the effect of a new set of strains of LAB, isolated from free-range Hy-Line Brown hens, as a potential candidate probiotic culture to reduce SE infection in neonatal turkey poults.

## 2. Materials and Methods

### 2.1. Salmonella Strain and Culture Conditions 

The organism used in all experiments was a poultry isolate of *Salmonella enterica* serovar Enteritidis (SE). Culture conditions followed the methodolgy descibed previously [15,16,17].

### 2.2. Isolation and Selection of Probiotic Candidates 

In the present study, ten 34-week-old free-range Hy-Line Brown hens were euthanized by CO_2_ inhalation. From each hen, briefly, cecal content was obtained, homogenized, serially diluted with 0.9% sterile saline solution, and plated on de Man Rogosa Sharpe (MRS) agar plates (MRS broth Catalog no. 288110, Becton Dickinson and Co., Sparks, MD, 21152, USA; Agar, Catalogue no. 211822, Becton Dickinson, Sparks, MD, 21152, USA). One single colony was obtained from each of the samples and then all isolates were identified by 16S rRNA sequence analyses (Microbial ID Inc., Newark, DE 19713, USA). Aliquots of 1 mL of each bacterial strain were maintained in 50% glycerol frozen stocks at −80 °C. Selected candidates were routinely cultured at 37 °C under microaerophilic conditions in MRS broth. Each isolate was passed three times, at every 8 h in MRS broth. Then bacteria were washed three times (Sorval RT7, Woburn, MA, USA), resuspended in sterile PBS and adjusted to an optical density (OD_600_) of 0.8–0.9 (Spectronic 20D+, Thermo Spectronic, Somerville, MA, USA). Each isolate was tested for Gram stain affinity, catalase, and oxidase production [26,27]. Aliquots of the combined culture containing ten selected LAB isolates were grown on MRS and used in the present study. This probiotic was diluted in sterile saline to 4 × 10^6^ cfu/mL for oral gavage and confirmed by spread plating on MRS. The morphological characteristics and identification of LAB lactic acid bacteria probiotic candidates are summarized in Table 1.

### 2.3. Experimental Design

Two independent experiments were conducted to evaluate the effect of the LAB probiotic candidate for treating *Salmonella enterica* serovar Enteritidis infection in turkey poults. In each experiment, sixty-one day-of-hatch female turkey poults were obtained from a local hatchery. Poults did not receive any vaccines, antibiotics, probiotics, or treatments at the hatchery. In both experiments, poults were challenged via oral gavage (0.25 mL) with 10^4^ cfu/poult of SE and randomly allocated to one of two groups (*n* = 30 poults): (1) the positive control group and (2) the probiotic treated group. Heated brooder batteries were used for housing each group separately and poults were permitted ad libitum access to water and unmedicated turkey starter feed, formulated to meet National Research Council-recommended levels of critical nutrients [28]. 1 h after the SE challenge, birds were orally gavaged with (0.25 mL) 10^6^ cfu/poult of LAB probiotic culture or (0.25 mL) PBS to control birds. A total of 24 h post-treatment, all poults were euthanized by CO_2_ inhalation. Twenty poults were socked in a bleach solution (3/4 cup of bleach to 1 gallon of cool water), and then the skin from the abdominal cavity was removed. With forceps and scissors flamed in an 80% ethanol solution, a 2 cm section was made and then the breast was removed. The ceca and cecal tonsils (CCT) were collected aseptically for SE recovery as described below. Poults used in all experiments were cared for using procedures approved by the University of Arkansas Institutional Animal Care and Use Committee (IACUC) protocol number 19021. Upon arrival, ten extra day-of-hatch poults were euthanized by CO_2_ asphyxiation. Ceca-cecal tonsils, liver, yolk sac, and spleen were aseptically cultured in a tetrathionate enrichment broth (Catalog no. 210420, Becton Dickinson, Sparks, MD, USA). Enriched samples were confirmed negative for *Salmonella* by streak plating the samples on Xylose Lysine Tergitol-4 (XLT-4, Catalog no. 223410, BD Difco™, Sparks, MD, USA) selective media.

### 2.4. Salmonella Recovery

The CCT collected in both experiments (*n* = 20 poults/group) were homogenized and diluted with saline (1:4 w/v). CCT homogenate samples were diluted by ten-fold serial dilutions and 100 µL were plated on brilliant green agar (BGA, Catalog no. 70134, Sigma St. Louis, MO, USA) plates containing 25 µg/mL novobiocin (NO, catalog no. N-1628, Sigma St. Louis, MO, USA) and 20 µg/mL of nalidixic acid (NA, catalog no. N-4382, Sigma, St. Louis, MO, USA), incubated at 37 °C for 24 h, then enumerated for total *Salmonella enterica* serovar Enteritidis cfu. Following plating to enumerate total *Salmonella enterica* serovar Enteritidis, the CCT homogenate samples were enriched with tetrathionate enrichment broth, 1× final dilution, and further incubated at 37 °C for 24 h. Enrichment samples were streaked onto XLT-4) selective media for confirmation of *Salmonella* presence. Plates that were negative on the enumeration method but were positive on enrichment were considered as 500 cfu/g as the limit of detection for SE viability.

### 2.5. Statistical Analysis

Log_10_ cfu/g of *Salmonella enterica* serovar Enteritidis in cecal contents were subjected to one-way analysis of variance as a completely randomized design, using the General Linear Models procedure of SAS [29]. Significant differences amongst the means were determined by Duncan's multiple range test at *p* ≤ 0.05. Enrichment data were expressed as positive/total poults (%), and the percent recovery of *Salmonella enterica* serovar Enteritidis was compared using the chi-squared test of independence, testing all possible combinations to determine the significance (*p* ≤ 0.05) for these studies [30].

## 3. Results and Discussion

The poultry industry is the fastest growing animal industry and is expected to grow continuously as demand for meat and eggs is accelerating due to growing populations, increasing incomes, and urbanization [31]. However, salmonellosis remains one of the most comprehensive foodborne diseases that can be transmitted to humans through animal and plant products [32,33,34].

Probiotics have been evaluated as a promising alternative to AGP’s by several scientists. However, the mechanisms of action for improving performance and health remains poorly understood [35,36,37,38]. Several published studies indicate that probiotics can modulate pro-inflammatory and anti-inflammatory cytokines [39,40] and exert anti-oxidant properties [41,42,43,44], as well as enhance intestinal integrity [45], innate immunity [46,47,48], and humoral immunity [44,49,50]

*Salmonella* Enteritidis infection in chickens has been shown to increase of heterophils-to-lymphocyte ratio [51,52]. Recent studies published by our laboratory have also shown marked heterophilia and lymphopenia in chickens challenged with SE. However, these hematological changes were prevented in chickens that received FloraMax^®^-B11 1 h after an SE challenge, as well as a reduction in intestinal colonization by SE and reduction in intestinal permeability of Fluorescein isothiocyanate-dextran (FITC-d) [53]. *Salmonella* infections are associated with disruption of the tight junctions (TJ) and inflammation [54]. Hence, gut integrity is escential to maintain optimal health [55,56,57,58]. Furthermore, SE endotoxins activate aldose reductase and nuclear factor (NF-κB) inducing inflammation [59,60,61,62]. The increase in oxidative stress has been associated with an increase in gut permeability [63,64,65]. Interestingly, microarray analysis with FloraMax^®^-B11 in broiler chickens challenged with SE showed a significant reduction in intestinal gene expression associated with the NF-κB complex and AR [66]. Hence, our studies suggest that the probiotic preserved intestinal integrity, helping to maintain innate defense mechanisms of the gastrointestinal tract [67,68,69]. These results are in agreement with numerous studies demonstrating that probiotics prevent *Salmonella* translocation, suppressed the oxidant-induced intestinal permeability, and improve intestinal barrier function [44,70,71,72,73]. In the present study, we evaluated the effects of a LAB probiotic candidate. Table 2 shows the results of the evaluation of the LAB probiotic candidate on CCT colonization of SE in turkey poults at 24 h post challenge. In Experiment 1, the probiotic significantly reduced the incidence of ceca/cecal tonsil SE following treatment, from 80% in control poults to 35% in treated poults (*p* ≤ 0.05). Administration of the probiotic reduced SE recovered 24 h following treatment by 2.71 log_10_ cfu/g, as compared with control. A similar trend was observed in Experiment 2, where the probiotic reduced the incidence of SE from 75% in controls to 25% in treated poults; a 3.03 log_10_ SE reduction in treated poults when compared with the control group. This data suggests that administration of the probiotic 1 h post SE challenge significantly reduced the incidence of *Salmonella* recovery from CCT of neonatal poults as compared to untreated controls 24 h following treatment.

The results of the present study suggest that the administration of this LAB-based probiotic 1 h after an SE challenge can be useful in reducing the cecal colonization of this pathogen in neonatal poults, although further research is needed to elucidate the mechanism of this response. Because the 16sRNA identification revealed that several of the candidate strains are the same genus and species, further evaluation of the whole genome of the 10 candidate strains is currently under evaluation to elucidate if the strains are homologous.

## 4. Conclusions

In both trials, a significant reduction in the incidence and log_10_ cfu/g of SE were observed in poults treated with the probiotic when compared with control poults (*p* ≤ 0.05). Results showed that the application of this probiotic culture could reduce SE cecal colonization in day-of-hatch turkey poults, although further research is needed to elucidate the mechanism of this response.

## Figures and Tables

**Table 1 animals-09-00696-t001:** Morphological characteristics and identification of lactic acid bacteria probiotic candidates ^1^

16s RNA Sequencing Microbial Identification.	Isolated Region	Gram Strain	Morphology	Catalase	Oxidase
*Lactobacillus johnsonii*	Ceca	+	Rods	−	−
*Weissella confusa*	Ceca	+	Irregular rods	−	−
*Lactobacillu salivarius*	Ceca	+	Rods	−	−
*Weissella confusa*	Ceca	+	Irregular rods	−	−
*Enterococcus faecium*	Ceca	+	Cocci (clusters)	−	−
*Weissella confusa*	Ceca	+	Irregular rods	−	−
*Lactobacillus johnsonii*	Ceca	+	Rods	−	−
*Lactobacillus johnsonii*	Ceca	+	Rods	−	−
*Lactobacillus johnsonii*	Ceca	+	Rods	−	−
*Lactobacillus salivarius*	Ceca	+	Rods	−	−

^1^ Symbols: (+), positive; (−), negative.

**Table 2 animals-09-00696-t002:** Evaluation of the lactic acid bacteria probiotic candidate on ceca-cecal tonsils (CCT) colonization of *Salmonella* Enteritidis in turkey poults at 24 h post challenge.

Treatments	Log_10_ cfu/g CCT ^1^	CCT incidence (%) ^2^
Experiment 1		
Positive *Salmonella* Enteritidis control (PBS)	6.16 ± 0.38 ^a^	16/20 (80%)
*Salmonella* Enteritidis with Probiotic culture	3.45 ± 0.56 ^b^	7/20 (35%) *
Experiment 2		
Positive *Salmonella* Enteritidis control (PBS)	5.17 ± 0.24 ^a^	15/20 (75%)
*Salmonella* Enteritidis with Probiotic culture	2.15 ± 0.90 ^b^	5/20 (25%) *

Turkey poults were orally gavaged with 10^4^ cfu of *Salmonella* Enteritidis at 1 d old. 1 h later, they were orally gavaged with the probiotic candidate at 10^6^ cfu. Samples were collected 24 h later. ^1^ Data expressed in Log_10_ cfu /g of tissue. Mean ± SE. ^a-b^ Values within treatment columns for each treatment with different superscripts differ significantly (*p* < 0.05). ^2^ Data expressed as positive poults to SE/total poults culture (%). * Indicates significantly different (*p* ≤ 0.05).

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
