# Peer review of "Isolation and Identification of Lactic Acid Bacteria Probiotic Culture Candidates for the Treatment of Salmonella enterica Serovar Enteritidis in Neonatal Turkey Poults"

_animals, 2019, doi:10.3390/ani9090696_

Round 1

Reviewer 1 Report

This is an interesting paper and novel findings have been reported. The experiment has been property described and results well presented and discussed. The language is also fine. Thus, based on my opinion the paper merits the acceptance in Animals. 

Author Response

Answers to Reviewer 1

This is an interesting paper and novel findings have been reported. The experiment has been property described and results well presented and discussed. The language is also fine. Thus, based on my opinion the paper merits the acceptance in Animals. 

We thank you very much for the time you have spent on reviewing our manuscript and your kind comments about the manuscript.

Reviewer 2 Report

Manuscript ID: animals-576198-peer-review-v1

Title: Isolation and identification of lactic acid bacteria probiotic culture candidates for the treatment of Salmonella enterica serovar Enteritidis in neonatal turkey poults

In this work the authors aimed at using poultry-isolated Lactobacillus spp.-based probiotic (LAB) as an alternative for antibiotics to reduce the burden of infection with Salmonella enteritis serovar Enteritidis in turkey poults. A combination of 10 LAB bacteria isolates was successfully used to reduce S. Enteritidis when shortly administered to turkey poults after a moderate challenge. The reported results are interesting, but some comments below need to be addressed.

Specific Comments

Line 45-48: The way these results are reported needs to be changed – see below comments for Lines 181-185 for suggestions. The introduction does not refer to any specific studies that used probiotics for Salmonella treatments. Please list some of these studies whether in poultry or non-poultry species to inform the readers with the previous efforts and why your study is different. Line 88: Change “back yard” to “backyard” Lines 97-99: “Aliquots of the combined culture containing ten selected lactic acid bacteria isolates were grown on MRS and used in the present study.” What is the proportion used from each of the 10 LAB isolates that constituted the combined culture? Is it 10% of each? Lines 102-103: “The morphological characteristics and identification of lactic acid bacteria probiotic candidates are summarized in Table 1.” The table shows 3 isolates of Weissella confusa, 2 isolates of Lactobacillus salivarius, and 4 isolates of Lactobacillus johnsonii. Are the isolates from each type different? and if so, how this was verified? If the isolates from each type were identical, why not only one representative isolate was chosen to include in the LAB mix? Using a fewer number of isolates would be more feasible and manageable! Ideally, I would recommend using only one isolate from each of the four unique LAB identified with 25% proportion each. Lines 110-111: “Poults were randomly assigned to treatment groups, then challenged 110 via oral gavage (0.25 mL) with approximately cfu/chick of S. Enteritidis….” Is this the standard challenge dose for S. Enteritidis? Any reference? With other Salmonella strains, we would use 10^6 to 10^8 cfu/bird. Line 127: Change “24” to “24 h”. Line 149: “…positive/total chickens…” - Do you mean “…positive/total poults…” Line 190: “…from 12 chickens.” – Do you mean “…from 12 poults.” Lines 113-114: “groups. Twenty-four hours post-treatment, all poults were euthanized….” Would not it be useful to add another later time point for sample collection post-treatment/challenge to measure the long-term effect of the treatment, e.g. 3 days post-treatment? Using only 1-day post-treatment time point may not completely show the end result from the probiotic/salmonella interaction in the GI tract. Lines 181-185: “However, significant increases in the proportion of Proteobacteria were observed in control poults when compared to poults that received the probiotic culture. At the Class level, it was interesting to observe that control poults had an increase in Gammaproteobacteria when compared with poults treated with the probiotic (Table 3).” First: Reporting of these results need to be changed: Normally the increase (or decrease) occurs because of the effect of a treatment. Since the control group had no treatment, such an effect (increase or decrease) is not expected to happen. So, the results should be expressed the other way around. E.g., “Significant decreases in the proportion of Proteobacteria were observed in poults that received the probiotic culture when compared with control poults." And “Poults treated with the probiotic had a decrease in Gammaproteobacteria when compared with control poults” Same changes need to be done in the abstract. Second: How do you explain these results? Look up in the literature! There might be a competition on resources between probiotic bacteria and Proteobacteria/ Gammaproteobacteria in the GI tract. This is something to consider!!! The probiotic dose may have been too high!!! A lower dose may be considered in a new experiment to see the difference.

<<END>>

Author Response

Answers to Reviewer 2

In this work the authors aimed at using poultry-isolated Lactobacillus spp.-based probiotic (LAB) as an alternative for antibiotics to reduce the burden of infection with Salmonella enteritis serovar Enteritidis in turkey poults. A combination of 10 LAB bacteria isolates was successfully used to reduce S. Enteritidis when shortly administered to turkey poults after a moderate challenge. The reported results are interesting, but some comments below need to be addressed.

We thank you very much for the time you have spent on reviewing our manuscript. We have given full consideration to your comments and the manuscript that has been carefully revised and modified accordingly. Taking in consideration the comments of the reviewers, the manuscript has been significantly modified in all sections, and even change the scope of the manuscript, from full article to short communication of preliminary results.  Please refer to the point-by-point reply to the Reviewers’ comments. 

Specific Comments

Line 45-48: The way these results are reported needs to be changed – see below comments for

Suggestion accepted.  this manuscript.  Thank you.

Lines 181-185 for suggestions. The introduction does not refer to any specific studies that used probiotics for Salmonella treatments. Please list some of these studies whether in poultry or non-poultry species to inform the readers with the previous efforts and why your study is different.

Suggestion accepted, the introduction has been improved and new references have been incorporated accordingly.  Thank you.

Line 88: Change “back yard” to “backyard”

Suggestion accepted.  A better description has been included this manuscript (ten 34 week-old free-range Hy-Line Brown hens).  Thank you.

Lines 97-99: “Aliquots of the combined culture containing ten selected lactic acid bacteria isolates were grown on MRS and used in the present study.” What is the proportion used from each of the 10 LAB isolates that constituted the combined culture? Is it 10% of each?

Suggestion accepted.  A better description of the preparation of the probiotic culture has been incorporated.  Thank you.

 Lines 102-103: “The morphological characteristics and identification of lactic acid bacteria probiotic candidates are summarized in Table 1.” The table shows 3 isolates of Weissella confusa, 2 isolates of Lactobacillus salivarius, and 4 isolates of Lactobacillus johnsonii. Are the isolates from each type different? and if so, how this was verified? If the isolates from each type were identical, why not only one representative isolate was chosen to include in the LAB mix? Using a fewer number of isolates would be more feasible and manageable! Ideally, I would recommend using only one isolate from each of the four unique LAB identified with 25% proportion each.

You are right about your statement.  In the present study, ten 34-week-old free-range Hy-Line Brown hens were euthanized by CO2 inhalation.  From each hen, briefly, cecal content was obtained, homogenized, serially diluted with 0.9% sterile saline solution and plated on de Man Rogosa Sharpe (MRS) agar plates.  To try to keep the culture simple, one single colony was obtained from each of the samples and all then isolates were identified by 16S rRNA sequence analyses. Aliquots of 1 mL of each bacterial strain were maintained in 50% glycerol frozen stocks at -80°C. Selected candidates were routinely cultured at 37°C under microaerophilic conditions in MRS broth. Aliquots of the combined culture containing ten selected LAB isolates were grown on MRS and used in the present study. This probiotic was diluted in sterile saline to an expected concentration of 4 x 106 cfu/mL for oral gavage to poults in these studies. Actual colony-forming units administered per poult from each experiment are reported, which were determined retrospectively from spread plating on MRS.  Because the 16sRNA identification revealed that several of the candidate strains are the same genus and specie, further evaluation of the whole genome of the 10 candidate strains is currently under evaluation to elucidate if the strains are homologous.

Lines 110-111: “Poults were randomly assigned to treatment groups, then challenged 110 via oral gavage (0.25 mL) with approximately cfu/chick of S. Enteritidis….” Is this the standard challenge dose for S. Enteritidis? Any reference? With other Salmonella strains, we would use 10^6 to 10^8 cfu/bird.

The justification of the SE challenge dose and time of bacterial colonization has been incorporated in the introduction with the proper references that now are in the reference section.  Thank you.

Line 127: Change “24” to “24 h”.

Suggestion accepted.  A better description has been included this manuscript (ten 34 week-old free-range Hy-Line Brown hens).  Thank you.

Line 149: “…positive/total chickens…” - Do you mean “…positive/total poults…”

Yes, clarification has been made in the text, thank you.

 Line 190: “…from 12 chickens.” – Do you mean “…from 12 poults.”

Yes, clarification has been made in the text, thank you.

Lines 113-114: “groups. Twenty-four hours post-treatment, all poults were euthanized….” Would not it be useful to add another later time point for sample collection post-treatment/challenge to measure the long-term effect of the treatment, e.g. 3 days post-treatment? Using only 1-day post-treatment time point may not completely show the end result from the probiotic/salmonella interaction in the GI tract.

Thank you for your comment.  We have included over 10 manuscripts that support the challenge model and culture at 24 h post challenge. 

Lines 181-185: “However, significant increases in the proportion of Proteobacteria were observed in control poults when compared to poults that received the probiotic culture. At the Class level, it was interesting to observe that control poults had an increase in Gammaproteobacteria when compared with poults treated with the probiotic (Table 3).” First: Reporting of these results need to be changed: Normally the increase (or decrease) occurs because of the effect of a treatment. Since the control group had no treatment, such an effect (increase or decrease) is not expected to happen. So, the results should be expressed the other way around. E.g., “Significant decreases in the proportion of Proteobacteria were observed in poults that received the probiotic culture when compared with control poults." And “Poults treated with the probiotic had a decrease in Gammaproteobacteria when compared with control poults” Same changes need to be done in the abstract. Second: How do you explain these results? Look up in the literature! There might be a competition on resources between probiotic bacteria and Proteobacteria/ Gammaproteobacteria in the GI tract. This is something to consider!!! The probiotic dose may have been too high!!! A lower dose may be considered in a new experiment to see the difference.

We have to agree with you and reviewer 3.  You both are correct in your comments regarding microbiota analysis and results.  Hence, we have decided to eliminate all the results of microbiota from this manuscript and that is one of the reasons we prefer to present these preliminary results as a short communication paper.  Thank you.

Reviewer 3 Report

Lines 6-10 and 205-212: Considering the small quantity of data presented in the manuscript (2 data tables), thirteen authors seems excessive for this manuscript. Please review the author contributions and consider limiting authorship to those who had intellectual input to the manuscript. Lines 26-27: I disagree with the sentence “Salmonella control has been traditionally been carried out with the use of antibiotics”. As written, it is misleading. Antibiotics may be part of a control strategy, but is not the only method used. Vaccination, surveillance and biosecurity are much more commonly used. The sentence should be rewritten, or omitted. Lines 32-50: The abstract has no background to the question or hypothesis being tested, and should be rewritten to include sufficient background for readers to comprehend the rationale for the current study. Lines 54-69: The introduction lacks sufficient background about Salmonella as a foodborne pathogen. Line 61: Most Salmonella in poultry are commensals of the distal gastrointestinal tract, and asymptomatically colonize. Line 65: Should read “lactic acid-producing bacteria (LAB)”. Line 67: “Microflora” should be changed to “microbiota”. Lines 72-79: The authors’ need to clarify whether nalidixic acid and novobiocin resistance are chromosomal or plasmid based, and whether they cultured the isolate in the presence of these antibiotics while preparing the inoculum. Line 88: The term hens is ambiguous. Please clarify the probiotics were isolated from turkey or chicken hens. Line 89: Please clarify how you separated intestinal epithelium for isolation of the probiotic strains. Lines 90-93: Please clarify the temperature and growth conditions (e.g., aerobic, microaerophilic or anaerobic) for culture of probiotic candidates on MRS media. Line 94: Why culture at 37⁰C when the core body temperature of poultry is 42⁰C? Please list the vendor of MRS broth. Lines 100 and 101: Chicks and poults are used on these lines. Please clarify which term is correct. Lines 104-118: The experimental design is confusing and lacks sufficient detail to reproduce these studies. There is confusion in the number of animals used for 16S analysis and Salmonella Did the poults receive any vaccines, antibiotics, probiotics or treatments at the hatchery? Were the poults free of Salmonella when they arrived to your animal facility? At what age were the poults in experiments 1 and 2 inoculated with Enteritidis (SE)? The number of animals per treatment group is not clear. On lines 111-112, you indicate n=30 per pen for experiment 1, but data from 20 poults is presented in table 1. Which is correct? If 30 were inoculated, what happened to the data from the additional 10 poults in table 1? How many animals were used for experiment 2? There is no clear indication. Were the day of hatch poults used in experiments 1 and 2 group housed before SE inoculation or segregated upon arrival? If they were physically segregated at their arrival and not allowed to live together, the 16S analysis will likely be different due to the “pen effect”. Were all the poults experiments 1 and 2 in a treatment group housed together or in different pens? Splitting the treatment groups into different rooms will increase the degrees of freedom and lessen the pen effect. How many poults were used for 16S analysis? Line 116 indicates n=5 per treatment, but line 130 indicates n=6. Please clarify this discrepancy. Why was 24h post-inoculation or treatment selected as the end point for SE enumeration and microbiota analysis? Considering the age of the poults and the immaturity of their microbiota, it’s very difficult to argue that any differences in phyla are valid and not due to a pen effect. Lines 121-128. The method for Salmonella recovery and enumeration requires additional details. Please describe the aseptic technique used to harvest the cecum and cecal tonsil (CCT) from the inoculated poults. It is vital that the poults are free of naladixic acid and novobiocin resistant Salmonella before challenge. Were they free of this? If so, please indicate how you tested for this. If not, interpreting the data will be more challenging. Please describe how you homogenized the CCT. Please indicate your criteria for a SE negative or SE positive poult (e.g., negative if no SE were recovered by direct plating of CCT or after enrichment). What is the limit of detection (cfu/g of tissue and/or cecal contents) for this SE isolate? For statistical analysis, what value do you assign samples in table 2 that were negative for SE by direct plating, but positive by enrichment? Line 130 – please clarify the number of samples/treatment group used for 16S analysis. How did you determine that n=5 or 6 is adequate to statistically determine community differences between the treatment groups? Please indicate this in the discussion. Table 1 – It is unclear why 4 isolates of johnsonii, 3 isolates of W. confusa and 2 isolates of L. salivarius were used in the probiotic cocktail. Were they isolated from different hens? If so, they need to be identified as different strains. Did you demonstrate that any of these isolates produce lactate in vitro? Lines 154-162 – This section should be in the introduction, not results and discussion. Table 2 – The treatments should be written clearly (e.g., SE without probiotics and SE with probiotics) and the number of animals used is VERY confusing. All animals should have been cultured (direct plating and enrichment) and all data should be reported. Line 190 states that n=12 were sampled for cfu/g of CCT, but incidence is reported as n=20. What happened to the other 8 samples for direct plating? Log10 cfu/g CCT is confusing and should be renamed to “Direct plating log10 cfu/g of CCT”. CCT incidence (%) is confusing and should be renamed to “SE positive by enrichment” Please clarify whether 20 or 30 poults (listed in experimental design) were used to generate enrichment data for this table. If data are from 20, where are data from the remaining 10? Table 3 and lines 178-185 – Why would you expect any differences in the microbiota of a 1 day old poult? Their microbiota are extremely immature and deficient of Firmicutes at day 1 because facultative anaerobes haven’t yet consumed enough oxygen to allow Firmicutes to become vegetative. In my experience, it takes up to 14 days before a stable microbiota is established in the ceca of poults. I feel these data should be removed from the manuscript for the following reasons: a) The number of samples is unclear (n=5 or n=6?), b) the number of samples are likely inadequate to have the statistical power to demonstrate any compositional differences and c) the pen effect (acquiring microbes from different pens) is likely driving more change than the treatment. 16S data should be reported as a figure (percent relative abundance of the phyla). Additional analysis (e.g., alpha and beta diversity, PCA analysis), beyond presenting select phyla and classes in a table, are required. Lines 163-177 and 196-203 – There is very little discussion of the results in these sections, which needs to be changed in the revised manuscript. The authors’ need to justify why 24h was an adequate timepoint to assess the effect of LAB probiotic administration on SE colonization. A very similar study was performed in day of hatch chicks (https://academic.oup.com/ps/article/86/8/1662/1521579) and they saw dramatic changes in the number of SE recovered from LAB treated chicks. Studies like this must be included in your discussion. Comparing your results to literature describing the inhibitory effect of LAB co-culture on SE in vitro or in vivo (e.g., https://onlinelibrary.wiley.com/doi/full/10.1111/jfs.12141, https://www.ncbi.nlm.nih.gov/pmc/articles/PMC3984083/ https://www.ncbi.nlm.nih.gov/pubmed/21983108, https://journals.plos.org/plosone/article?id=10.1371/journal.pone.0147630 ), and benefits of LAB and bacteriocins as alternatives to antibiotics (https://www.frontiersin.org/articles/10.3389/fmicb.2019.00057/full) should be included in the revised results and discussion.

Author Response

Answers to Reviewer 3

We thank you very much for the time you have spent on reviewing our manuscript. We have given full consideration to your comments and the manuscript that has been carefully revised and modified accordingly. Taking in consideration the comments of the reviewers, the manuscript has been significantly modified in all sections, and even change the scope of the manuscript, from full article to short communication of preliminary results.  Please refer to the point-by-point reply to the Reviewers’ comments

Lines 6-10 and 205-212: Considering the small quantity of data presented in the manuscript (2 data tables), thirteen authors seems excessive for this manuscript. Please review the author contributions and consider limiting authorship to those who had intellectual input to the manuscript.

Suggestion accepted.  Four co-authors have been removed from this manuscript.  Thank you.

Lines 26-27: I disagree with the sentence “Salmonella control has been traditionally been carried out with the use of antibiotics”. As written, it is misleading. Antibiotics may be part of a control strategy, but is not the only method used. Vaccination, surveillance and biosecurity are much more commonly used. The sentence should be rewritten, or omitted. .

Suggestion accepted.  Sentence has been removed.  Thank you.

Lines 32-50: The abstract has no background to the question or hypothesis being tested, and should be rewritten to include sufficient background for readers to comprehend the rationale for the current study.

We agree with you, unfortunately, there is a restriction of in the number of words for the abstract, according to the guidelines for authors. The abstract should be a total of about 200 words maximum.  The abstract has been re-written.  Thank you.

Lines 54-69: The introduction lacks sufficient background about Salmonella as a foodborne pathogen.

Suggestion accepted, the introduction has been improved and new references have been incorporated accordingly.  Thank you.

Line 61: Most Salmonella in poultry are commensals of the distal gastrointestinal tract, and asymptomatically colonize.

Suggestion accepted, that section was removed, thank you.

Line 65: Should read “lactic acid-producing bacteria (LAB)”.

Suggestion accepted, thank you.

 Line 67: “Microflora” should be changed to “microbiota”.

Suggestion accepted, thank you.

 Lines 72-79: The authors’ need to clarify whether nalidixic acid and novobiocin resistance are chromosomal or plasmid based, and whether they cultured the isolate in the presence of these antibiotics while preparing the inoculum.

This strain of SE has been utilized for over 30 years in our laboratory and in multiple publications.  Until today, we have not determined whether nalidixic acid and novobiocin resistance are chromosomal or plasmid based.  The culture of the isolate is without these antibiotics while preparing the inoculum.  This statement has been incorporated in the text, thank you.

Line 88: The term hens is ambiguous. Please clarify the probiotics were isolated from turkey or chicken hens.

Suggestion accepted, text has been modified, thank you.

Line 89: Please clarify how you separated intestinal epithelium for isolation of the probiotic strains.

Suggestion accepted, text has been modified, thank you.

 Lines 90-93: Please clarify the temperature and growth conditions (e.g., aerobic, microaerophilic or anaerobic) for culture of probiotic candidates on MRS media.

Selected candidates were routinely cultured at 37°C under microaerophilic conditions in MRS broth as indicated in the text, thank you.

Line 94: Why culture at 37⁰C when the core body temperature of poultry is 42⁰C? Please list the vendor of MRS broth.

We use standard temperature of 37⁰C in all our incubators.  Most poultry commensal or pathogenic bacteria can growth well on the range of 37⁰C to 42⁰C. Man Rogosa Sharpe (MRS) agar plates were made by combining according concentrations of MRS broth and Agar flakes.  This clarification and the vendors have been including in the text, thank you.

Lines 100 and 101: Chicks and poults are used on these lines. Please clarify which term is correct.

Very good catch, clarification has been made, thank you!

Lines 104-118: The experimental design is confusing and lacks sufficient detail to reproduce these studies. There is confusion in the number of animals used for 16S analysis and Salmonella Did the poults receive any vaccines, antibiotics, probiotics or treatments at the hatchery? Were the poults free of Salmonella when they arrived to your animal facility? At what age were the poults in experiments 1 and 2 inoculated with Enteritidis (SE)? The number of animals per treatment group is not clear.

The experimental section has been rewritten to clarify the methodology such as the age of inoculation with SE and probiotic, the number of poults utilized during the culture procedure in Experiments 1 and 2, , as well as the number of samples collected for microbiome analysis only in experiment 2.  Poults did not receive any treatment at the hatchery, and this statement has been incorporated in the text too.  Confirmation that the poults were Salmonella free when they arrived to our facility has also been incorporated in the text, thank you very much.

On lines 111-112, you indicate n=30 per pen for experiment 1, but data from 20 poults is presented in table 1. Which is correct? If 30 were inoculated, what happened to the data from the additional 10 poults in table 1? How many animals were used for experiment 2? There is no clear indication. Were the day of hatch poults used in experiments 1 and 2 group housed before SE inoculation or segregated upon arrival? If they were physically segregated at their arrival and not allowed to live together, the 16S analysis will likely be different due to the “pen effect”. Were all the poults experiments 1 and 2 in a treatment group housed together or in different pens? Splitting the treatment groups into different rooms will increase the degrees of freedom and lessen the pen effect. How many poults were used for 16S analysis?

Clarification of all your questions have been made accordingly in the experimental design.  Thank you.

Two independent experiments were conducted to evaluate the effect of the LAB probiotic candidate for treating S. Enteritidis infection in turkey poults.  In each experiment, sixty-one day-of-hatch female turkey poults were obtained from a local hatchery.  Poults did not receive any vaccines, antibiotics, probiotics or treatments at the hatchery.  In both experiments, poults were challenged via oral gavage (0.25 mL) with approximately 104 cfu/poult of S. Enteritidis and randomly allocated to one of two groups (n=30 poults): 1) positive control group; and 2) Probiotic treated group. Heated brooder batteries were used for housing each group separately and poults were allowed ad libitum access to water and unmedicated turkey starter feed, formulated to meet or exceed National Research Council-recommended levels of critical nutrients [9], for the duration of the experiment. One hour following Salmonella challenge, poults were treated with 106 cfu/poult of probiotic culture via oral gavage (0.25 mL) or PBS to control groups. Twenty-four hours post-treatment, all poults were euthanized by CO2 inhalation and ceca and cecal tonsils (CCT) from twenty poults were collected aseptically for S. Enteritidis recovery as described below.  At this time, and only in Experiment 2, ceca samples from six poults were also collected to determine Phylum distribution, cumulative percent lowest common ancestor (LCA) and direct Class assignment in percentage as described belowPoults used in all experiments were cared for using procedures approved by the University of Arkansas Institutional Animal Care and Use Committee (IACUC) protocol number 19021.  Upon arrival, ten extra day-of-hatch poults were euthanized by CO2 asphyxiation. Ceca-cecal tonsils, liver, yolk sac and spleen were aseptically cultured in tetrathionate enrichment broth (Catalog no. 210420, Becton Dickinson, Sparks, MD). Enriched samples were confirmed negative for Salmonella by streak plating the samples on Xylose Lysine Tergitol-4 (XLT-4, Catalog no. 223410, BD Difco™) selective media.

Line 116 indicates n=5 per treatment, but line 130 indicates n=6. Please clarify this discrepancy. Why was 24h post-inoculation or treatment selected as the end point for SE enumeration and microbiota analysis? Considering the age of the poults and the immaturity of their microbiota, it’s very difficult to argue that any differences in phyla are valid and not due to a pen effect.

Clarification has been made (n=6), thank you.  The treatment of the probiotic one hour after SE challenge, is a model that has been utilized and extensively published by our laboratory.  The background on this model and the references have been incorporated in the introduction and reference section.  Regarding the age of the poults and the immaturity of their microbiota, you are right that is very difficult to argue differences, particularly, since we know that the ceca of young chickens are mainly dominated by the phylum Firmicutes, Proteobacteria, and Actinobacteria, whereas the relative abundance of Bacteriodetes increase with age and was detected only after 15 days in broiler chickens (Ranjitkar et al., 2016). The relevance of the findings observed in the present study at Phylum and Class level in such a short period of time utilizing the SE model, suggest that the effects of this probiotic candidate culture could be achieved through mechanism(s) that might involve the modulation of gut microbiota and their metabolic pathways. These statements have been added in the discussion section as well.  Thank you for your comments.

Lines 121-128. The method for Salmonella recovery and enumeration requires additional details. Please describe the aseptic technique used to harvest the cecum and cecal tonsil (CCT) from the inoculated poults. It is vital that the poults are free of naladixic acid and novobiocin resistant Salmonella before challenge. Were they free of this? If so, please indicate how you tested for this. If not, interpreting the data will be more challenging. Please describe how you homogenized the CCT. Please indicate your criteria for a SE negative or SE positive poult (e.g., negative if no SE were recovered by direct plating of CCT or after enrichment). What is the limit of detection (cfu/g of tissue and/or cecal contents) for this SE isolate? For statistical analysis, what value do you assign samples in table 2 that were negative for SE by direct plating, but positive by enrichment? Line 130 – please clarify the number of samples/treatment group used for 16S analysis. How did you determine that n=5 or 6 is adequate to statistically determine community differences between the treatment groups? Please indicate this in the discussion. Table 1 – It is unclear why 4 isolates of johnsonii, 3 isolates of W. confusa and 2 isolates of L. salivarius were used in the probiotic cocktail. Were they isolated from different hens? If so, they need to be identified as different strains. Did you demonstrate that any of these isolates produce lactate in vitro?

Specific details of the method for Salmonella recovery and enumeration has been incorporated.  Upon arrival, ten extra day-of-hatch poults were euthanized by CO2 asphyxiation. Ceca-cecal tonsils, liver, yolk sac and spleen were aseptically cultured in tetrathionate enrichment broth (Catalog no. 210420, Becton Dickinson, Sparks, MD). Enriched samples were confirmed negative for Salmonella by streak plating the samples on Xylose Lysine Tergitol-4 (XLT-4, Catalog no. 223410, BD Difco™) selective media.  This statement has been also included in the text.  Plates that were negative on the enumeration method but were positive on enrichment were consider as 500 CFU/gr as limit of detection for SE viability.  This sentence has also been added in the text.  The number of samples for 16S analysis has been corrected (n=6).  Individual isolates were obtained from ten 34 week-old free-range Hy-Line Brown hens.  Because the 16sRNA identification revealed that several of the candidate strains are the same genus and specie, further evaluation of the whole genome of the 10 candidate strains is currently under evaluation to elucidate if the strains are homologous.  This sentence is now in the discussion section.  We have not demonstrated that any of these isolate produce lactate in vitro.  Thank you.

Lines 154-162 – This section should be in the introduction, not results and discussion.

Suggestion accepted, thank you.

Table 2 – The treatments should be written clearly (e.g., SE without probiotics and SE with probiotics) and the number of animals used is VERY confusing. All animals should have been cultured (direct plating and enrichment) and all data should be reported. Line 190 states that n=12 were sampled for cfu/g of CCT, but incidence is reported as n=20. What happened to the other 8 samples for direct plating? Log10 cfu/g CCT is confusing and should be renamed to “Direct plating log10 cfu/g of CCT”. CCT incidence (%) is confusing and should be renamed to “SE positive by enrichment” Please clarify whether 20 or 30 poults (listed in experimental design) were used to generate enrichment data for this table. If data are from 20, where are data from the remaining 10?

Tables 2 and 3 legend for the treatments have been modified and the number of animals corrected accordingly, both in the material section and in Tables 2 and 3.  Clarification on the way data is expressed in Table 2 has also been included.  Only 20 poults in both experiments were cultured.  A total of 30 poults were included initially, to ensure to have at least 20 poults for the culture, which is the number that give us statistical support in our manuscripts for the last 30 years.  Thank you.

Table 3 and lines 178-185 – Why would you expect any differences in the microbiota of a 1 day old poult? Their microbiota are extremely immature and deficient of Firmicutes at day 1 because facultative anaerobes haven’t yet consumed enough oxygen to allow Firmicutes to become vegetative. In my experience, it takes up to 14 days before a stable microbiota is established in the ceca of poults. I feel these data should be removed from the manuscript for the following reasons: a) The number of samples is unclear (n=5 or n=6?), b) the number of samples are likely inadequate to have the statistical power to demonstrate any compositional differences and c) the pen effect (acquiring microbes from different pens) is likely driving more change than the treatment. 16S data should be reported as a figure (percent relative abundance of the phyla). Additional analysis (e.g., alpha and beta diversity, PCA analysis), beyond presenting select phyla and classes in a table, are required.

We must agree with you comments.  Not only the number of samples is inadequate to have the statistical power to demonstrate any compositional differences, but the data comes from one single experiment.  Furthermore, the time for evaluation of microbiome is too short.  We have decided to eliminate the microbiome analysis from this manuscript.  Thank you.

Lines 163-177 and 196-203 – There is very little discussion of the results in these sections, which needs to be changed in the revised manuscript. The authors’ need to justify why 24h was an adequate timepoint to assess the effect of LAB probiotic administration on SE colonization. A very similar study was performed in day of hatch chicks (https://academic.oup.com/ps/article/86/8/1662/1521579) and they saw dramatic changes in the number of SE recovered from LAB treated chicks. Studies like this must be included in your discussion. Comparing your results to literature describing the inhibitory effect of LAB co-culture on SE in vitro or in vivo (e.g., https://onlinelibrary.wiley.com/doi/full/10.1111/jfs.12141, https://www.ncbi.nlm.nih.gov/pmc/articles/PMC3984083/ https://www.ncbi.nlm.nih.gov/pubmed/21983108, https://journals.plos.org/plosone/article?id=10.1371/journal.pone.0147630 ), and benefits of LAB and bacteriocins as alternatives to antibiotics (https://www.frontiersin.org/articles/10.3389/fmicb.2019.00057/full) should be included in the revised results and discussion.

Justification to why 24h was an adequate timepoint to assess the effect of LAB probiotic administration on SE colonization has been included in the introduction with the support of several new references.

Round 2

Reviewer 3 Report

I thank the authors' for their well written rebuttal to my comments. The revised manuscript is suitable for publication.